# Environmental Impact Assessment of a Wharf Oil Spill Emergency on a River Water Source

Fei He [1], Jie Ma [1], Qiuying Lai [1], Jian Shui [1,2] and Weixin Li [1,*]

1   Nanjing Institute of Environmental Sciences, Ministry of Ecology and Environment, Nanjing 210042, China
2   College of Hydrology and Water Resources, Hohai University, Nanjing 210024, China
*   Correspondence: lwxnies@126.com

**Abstract:** In recent years, there have been frequent water pollution emergencies, which seriously threaten the environment of water supply sources and affect the safety and quality of the water supply. These emergencies have aroused concern from the public and the government and highlight the necessity of plans for the emergency treatment of the affected water sources. In this paper, a sudden pollution of a river drinking water source is used as the research object. A mathematical simulation method was used to investigate the water quality near and downstream of a wharf in the state of a sudden oil spill. The wharf is located 1.34 km upstream of the water intake position of this river water source. Based on our investigation, we have established a risk assessment method for an oil spill emergency pollution event. Our aim was to provide a basis for the assessment of the water intake quality and water safety status of a river-based water plant and to provide technical support for developing an effective treatment plan in the event of a sudden environmental pollution accident.

**Keywords:** oil spill emergency; water source; environmental impact assessment

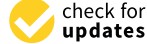



## 1. Introduction

In recent years, the number of sudden pollution incidents in China's urban water sources has become more numerous, and the environmental risks of water sources have been increasing [1–3]. The security of local water supplies is essential for both human life and regional economic growth since they provide a source of drinking water. The environmental concerns associated with China's water supplies are increasing as a result of the country's booming social economy and expanding urbanization, frequently resulting in pollution incidents affecting water sources [4]. To assure the reliability of the water supply, many academics have assessed the environmental risks associated with various water sources.

The environmental risk assessment of water sources at home and abroad is mainly categorized into water quality risk assessment and risk assessment, considering the influence of risk sources [4]. For the former, the focus of water quality risk assessment has shifted from conventional pollutants to the health risk assessment of emerging pollutants. For instance, J. Nawab et al. [5] and Ali Akbar Mohammadi et al. [6], respectively, conducted health risk analyses of water sources in the Malakand region of Pakistan and the Khorramabad region of Iran using heavy metals as the objects. Drugs served as the focus of health risk analyses carried out by C J Houtman et al. [7] and Feng et al. [8] on three water sources in the Netherlands and a Yangtze River water source in Chongqing, respectively. In the case of the latter, the authors found that the economic structure of developed countries exerts little impact on the environment, and such countries possess mature risk source control systems. While China is quickly developing its industry, which poses an increasing threat to the supply of safe drinking water, industrialized nations have built risk source management mechanisms that ensure that industrialization has little to no environmental impact. As a result, China is where most environmental risk assessment research into water supplies takes place that takes into account the consequences of risk sources.

According to a government investigative report, more than 1900 water contamination incidents were documented in China between 2003 and 2019, with automotive accidents accounting for 10.9% of those incidents. The primary cause of water contamination brought on by boat traffic accidents was diesel oil leakage. The natural equilibrium of the water was severely harmed by the oil slicks that developed after spills into lakes. As the primary location where commodities are loaded and unloaded from ships, busy wharf operations, ship entry and exit, or even inadequate safety precautions at the wharves, are prone to creating safety mishaps such as ship collisions and oil overturns, which result in accidents involving oil spills [4]. Thus, it is crucial to conduct a risk assessment of water sources taking into account the effects of oil spill incidents in order to safeguard the ecological environment of water sources and the security of water quality [9].

At present, dynamic simulation research of oil spill risk at home and abroad is mostly concentrated on offshore locations [10,11], and there are not many model-based studies on river oil spills and their risks and treatments [12]. Generally, a qualitative risk assessment of environmental factors is used for sudden pollution events in rivers [13–15]. However, to date, quantitative analysis and simulation studies on the risk of oil spillage on river-based water intakes have been rarely reported [12,16,17]. In our investigation, we set up a two-dimensional oil spill model that simulated the combined working conditions of different factors. In the model, the oil spill takes place 1.34 km upstream of the water intake and the risk analysis is based on this design. We also simulated the influence law and influence range of the oil spill on the water intake. All of these considerations in our model are of great significance to improving the emergency decision-making and management efficiency of sudden water pollution accidents that take place in rivers and other water bodies.

## 2. Methods

### 2.1. Study Area

Nanjing Dajian Wharf (118°32′ E, 24°52′ N) and Jiangning Binjiang Water intake (118°32′ E, 31°50′ N) are both located in the Tongjing Port Area of the Nanjing Jiangning Binjiang Economic Development Zone. They are located in the western part of the Jiangning District of Nanjing, the capital of Jiangsu Province and the central city of the Yangtze River Delta. They are close to the golden waterway of the Yangtze River, 25 km from the main city of Nanjing, about 40 km from the downstream Nanjing Xinshengwei Port Area, about 54 km from the upstream Wuhu Port, and about 432 km from Shanghai Port. The dominant wind direction throughout the year is northeast, with an average annual wind speed of 2.6 m/s.

### 2.2. Risk Identification

We first demonstrated that the wharf is mainly exposed to heavy transportation traffic, and it is not exposed to chemical, oil, or dangerous goods transportation. According to the analysis of dock accidents, the main sources of accidents involving oil spills are ship collisions or oil spilled by docked ships. Therefore, the spilled oil that affects water quality and the water at a water intake is usually the fuel oil used to power ships.

For the model, we used the actual situation of the wharf: an oil leakage event from a ship of the maximum permissible size docked at one berth. The survey data on the relationship between tonnage and fuel quantity of cargo ships in China is shown below.

Fuel load = maximum fuel capacity × actual load ratio of the fuel tank

The maximum carrying capacity of fuel for non-cruise ships is generally 8–12% of the total tonnage of the ship. The maximum acceptable tonnage for a dock is 5000 tons for dry bulk carriers. So, using the total tonnage of the ship as 5000 tons, we calculated that 8% of the total tonnage would be 400 tons. We used this value as the maximum carrying capacity of the fuel tank. The actual fuel load rate of the ship is about 10–20% when it arrives at the port. Therefore, according to the general ship setup with four fuel tanks, the single-tank fuel load capacity is about 10 tons when the ship arrives at the port.

Therefore, the occurrence of a large oil spill accident under the most unfavorable circumstances might happen in the following way: a docked ship collides during berthing or loading and unloading operations in the port, resulting in the rupture of a fuel tank, and 100% of the fuel oil in the tank is considered as having spilled into the water body. Thus, the maximum amount of fuel oil spilled into the river is about 10 t/time.

*2.3. Overview of the Forecast Method*

(1)   Goal:

To predict the impact of the design flow rate on the water body and water source under the design flow conditions of the dry period. According to the prediction results, the influence range and distance of oil spill film drift will be analyzed, focusing on the impact on the water source.

(2)   Scope of the prediction:

The study area will be from the confluence of the river 24 km upstream of the water intake to the section of the Yangtze River 3.4 km downstream of the water intake, a total distance of 27.4 km. The impact of fuel oil leaked from ships on water quality in this area will be predicted only.

(3)   Details:

The accident site was selected as the front of the proposed wharf, and fuel oil was used as the representative polluting substance. The risk assessment only analyzed the leakage of fuel oil after emergency measures were taken. The fuel oil leakage amount was 10 t. The specific prediction scheme is shown in the Table 1.

**Table 1.** The dock accident risk prediction scheme.

| Number | Water Condition | Discharge Conditions | Pollution Emissions | Predictors |
|---|---|---|---|---|
| 1 | Design flow of 90% low water | Accident emissions | 10 t | Petroleum |

*2.4. Methods of Forecast*

The area where the wharf is located on the Yangtze River is wide and shallow. Based on these characteristics and other parameters of our investigation, we used the average two-dimensional tidal flow model for water depth to simulate the water flow field in the evaluation area. Additionally, the oil slick drift extension model was used to simulate the oil slick diffusion process in the evaluation area.

2.4.1. Two-Dimensional Tidal Current Mathematical Model

(1)   Hydrodynamic model

Continuous equations:

$$\frac{\partial \xi}{\partial t} + \frac{\partial (Hu)}{\partial x} + \frac{\partial (Hv)}{\partial y} = 0 \tag{1}$$

Momentum equation:

$$\frac{\partial (Hu)}{\partial t} + \frac{\partial (Huu)}{\partial x} + \frac{\partial (Huv)}{\partial y} = fHv - gH\frac{\partial \xi}{\partial x} - \frac{gu\sqrt{u^2 + v^2}}{c^2} \tag{2}$$

$$\frac{\partial (Hv)}{\partial t} + \frac{\partial (Huv)}{\partial x} + \frac{\partial (Hvv)}{\partial y} = -fHu - gH\frac{\partial \xi}{\partial x} - \frac{gv\sqrt{u^2 + v^2}}{c^2} \tag{3}$$

In the formula, $x$ and $y$ refer to the longitudinal and horizontal coordinates, respectively; $u$ and $v$ refer to the average flow velocity component in the $x$ and $y$ directions,

respectively; $H$ refers to the full water depth (that is, the distance from the river bottom to the water surface); $\zeta$ refers to the water level; $f$ refers to the Kouhlet force coefficient; $g$ refers to the acceleration due to gravity; and $c$ refers to the Chézy coefficient calculated as: $c = \frac{1}{n}H^{\frac{1}{6}}$.

(2)　Definite condition

(i)　　Boundary conditions:

Shore boundary: The normal flow velocity of the shore boundary is zero: $\frac{\partial V}{\partial n} = 0$.

Water boundaries: The upstream and downstream boundaries use the tidal level process line, and the tidal level process is obtained according to the measured tidal level process.

(ii)　　Initial conditions:

$$u(x,y,0) = u_0(x,y); v(x,y,0) = v_0(x,y); z(x,y,0) = z_0(x,y)$$

(3)　Parameter selection

According to the characteristics of the Yangtze River and previous research results, the coefficient of roughness of the main trough of the Yangtze River is generally 0.018–0.022, and the coefficient of roughness of the river beach is generally 0.024–0.028.

### 2.4.2. Oil slick Drift Extension Model

The oil slick drift extension model we used in this prediction is called the oil particle model.

The oil spill model is a simulation method that regards oil spillage as a collection of a large number of particles of different masses, and it expresses the behavior of an oil spill in a water environment as the macroscopic movement of the particles [18–21]. The advection of the oil slick can be achieved by tracing the trajectory of the particles, which has a Lagrangian property and can be simulated by the Lagrangian method, usually using the method of additional volume parameters to simulate the characteristics of oil particles [22–24]. The volume parameter of an oil particle can be defined as:

$$V_i = \frac{\pi(d_i)^3}{6} \tag{4}$$

where $V_i$ is the volume of the $i$th oil particle, and $d_i$ is its diameter.

The percentage of the total volume of the oil slick, $f_i$, can be defined as:

$$f_i = \frac{\frac{\pi}{6}(d_i)^3}{\sum_{j=1}^{n} \frac{\pi}{6}(d_i)^3} \tag{5}$$

In the formula, $n$ is the total number of oil particles.

The characteristic volume of each oil particle can be defined as:

$$V_i = f_i \cdot V_0 \tag{6}$$

In the formula, $V_0$ is the initial volume of the oil spill. In this way, each oil particle represents a part of the spilled volume.

In this model, the oil spill can be thought of as a collection of a large number of small particles of the same mass. A set of spatial coordinates can be assigned to each particle. It is assumed that these small particles shift with the surrounding water and diffuse randomly. In other words, we simulated the shifting with deterministic methods and simulated the horizontal disturbance process with random walking.

(1) "Oil particles" drift displacement

At each time step, the displacement of the "oil particles" drift was superimposed by horizontal convection and perturbation, which can be calculated by the following formula:

$$S_x(\Delta t) = U_{cx}\Delta t + \Delta L cos\theta \tag{7}$$

$$S_y(\Delta t) = U_{cy}\Delta t + \Delta L sin\theta \tag{8}$$

$$\theta = 2\pi[R']_0^1 \tag{9}$$

in which $S_x(\Delta t)$ and $S_y(\Delta t)$ refer to the displacement of oil particles in a given direction, while $U_{cx}$ and $U_{cy}$ refer to the convection velocity of oil particles in that direction, and $[R']_0^1$ refers to a random number between 0~1 (independent of $[R]_0^1$).

(2) "Oil particles" are disturbed and diffused horizontally

The horizontal disturbance of "oil particles" mainly depends on turbulence and shear flow, which is random and simulated by the random walking method. The horizontal diffusion distance of "oil particles" can be determined by the following formula:

$$\Delta L = [R]_0^1\sqrt{12D\Delta t} \tag{10}$$

where $[R]_0^1$ refers to a random number between 0~1; $D$ refers to the horizontal diffusion coefficient, taken as 0.35 m$^2$/s; and $\Delta t$ refers to the time step.

### 2.4.3. Calculation Condition Selection

(1) Calculation range and meshing

According to the purpose of the study, the completeness of hydrological data, and the requirements of the model calculation, the calculation range was selected from 24 km upstream of the water intake to the 3.4 km downstream section of the water intake (in total, a 27.4 km river section), as shown in Figure 1. The mesh layout uses a rectangular mesh that produces a total of 134 (portrait) by 104 (horizontal) nodes, with a longitudinal mesh step of 150 m and a horizontal mesh step of 150 m. The river section uses a 1:10,000 underwater terrain contour map to read the river bottom elevation for each computed node.

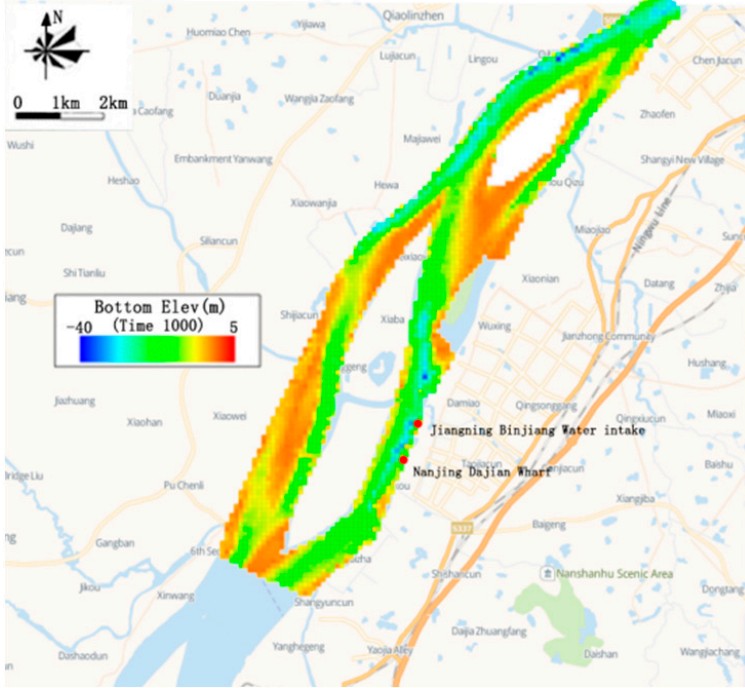

**Figure 1.** Calculation of the river area topography and grid layout.

(2) Calculate the selection of hydrological, meteorological, and water quality conditions

According to the hydrological and water quality design conditions, the impact prediction in the case of a material accidental discharge was carried out. Since the calculation area is in the tidal section, in a calculated tide pattern, the tide level and flow rate are changing from moment to moment, and the accidental discharge is a non-continuous discharge. The initial discharge time of the terminal pollutant in the accident situation is different, and the concentration field range formed is also different. According to the requirements of the corresponding national norms and procedures, from the perspective of safety, the average monthly flow rate with the lowest guarantee rate of 90% should be used as the design flow [25]. According to the minimum monthly average flow series measured by Datong Station across a number of years, the minimum monthly average flow rate with a 90% guarantee rate is 7580 m$^3$/s after frequency analysis. The wharf is located in the southwest direction of the water intake, so we choose a southwest wind, the most unfavorable wind, as the wind direction. And the wind speed is the instantaneous maximum wind speed of the southwest wind (5.0 m/s).

The influence of temperature on oil spill evaporation involves the evaporation rate and total evaporation ratio; the higher the temperature, the faster the oil evaporates; for the same oil, the total evaporation ratio is large at high temperatures and small at low temperatures [26–28], while temperature has less impact in the oil slick drift extension model [29–31]. The climate of the study area belongs to the northern subtropical monsoon climate zone, with an annual average temperature of 15 °C; so we choose 15 °C as the temperature.

(3) Model rate validation

The overall one-dimensional water flow and local two-dimensional tidal current mathematical model were used to simulate the Jiangsu section of the Yangtze River. According to the division of the two-dimensional simulation calculation area, the one-dimensional model provides the corresponding water flow boundary conditions for the two-dimensional model, and then the two-dimensional regional water quality coupling numerical simulation is carried out [32–34].

Datong Station with measured data was used as the test section, and the measured water level and the simulated calculated water level were compared. The data from 6:00 a.m. on 14 January 1979, to 6:00 a.m. on 18 January 1979 (96 h in total) were used for model calibration, and the measured water level and the simulated calculated water level were compared, respectively. The results show that the water surface line along the course is largely the same as the measured water surface line, and the percentage of the time period with the tidal level error not exceeding 20 cm in the total rate fixed period is between 80% and 98%. In general, the generalization of the area by the one-dimensional water flow model is largely reasonable, and the parameters selected broadly reflect the hydraulic characteristics of the river.

## 3. Results and Discussion

### 3.1. Calculation and Analysis of the Tidal Field

The tide in the river section where the wharf is located is an irregular semi-diurnal tide mixed type. The tide level rises and falls twice a day with high tide lasting for a short time and low tide lasting for a long time. This section of the river flows in one direction for most of the year, and there will be a reverse flow during the high tide and a small flow in the dry season, so the water flow conditions are more complicated. Since the evaluated water intake is located downstream of the oil spill location, this forecast mainly considers the impact of an oil spill on sensitive targets in the event of low tide. Through the calculation of our hydrodynamic model, the regional hydrodynamic flow field was simulated and evaluated. The distribution of the flow field at the time of the rise and rush is shown in Figure 2, and the distribution of the flow field at the time of the falling emergency is shown in Figure 3. The flow field map better reflects the flow movement of this section of the river. The mainstream largely flows in the direction of the main trough, the rising current goes

to the southwest, the falling current goes to the northeast, and the flow direction does not change much.

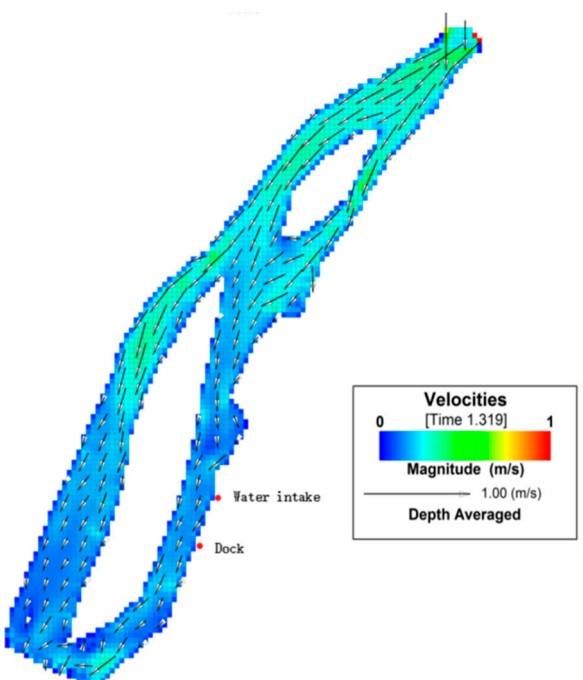

**Figure 2.** The distribution of the flow field of the calculation area at the time of rise and the emergency.

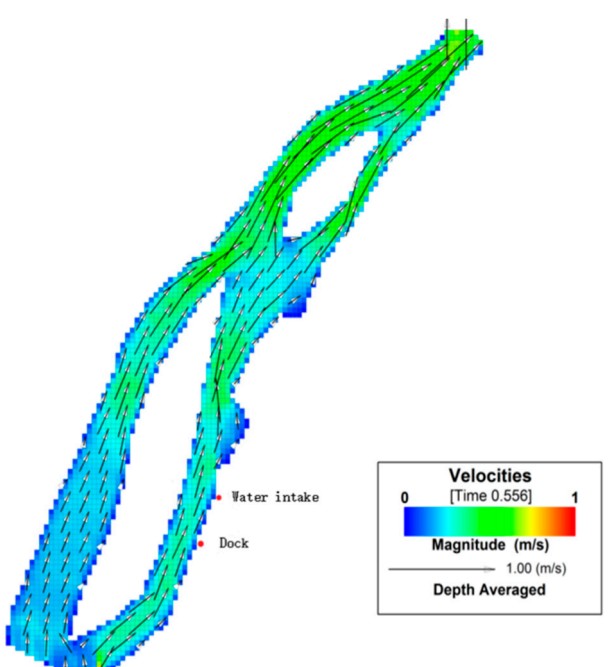

**Figure 3.** The flow field distribution map of the calculation area at the drop rush time.

Because the oil slick drift process mainly depends on the action of the wind field and flow field, it is necessary to study the influence of the wind field on the oil slick drift trajectory [35–38]. The predicted results of working conditions 1, 2, and 3 were compared when everything else was equal, and the effects of wind direction and speed on the drift and diffusion of the oil slick were analyzed (Table 2).

**Table 2.** The time of arrival and departure of the oil slick in different conditions.

| Conditions | Arrival Time | Departure Time |
|---|---|---|
| 1. Northeast wind, 2.6m/s | 40 min | 49 min |
| 2. Southwest wind, 2.6 m/s | 42 min | 51 min |
| 3. Southwest wind, 5.0 m/s | 46 min | 66 min |

Comparing the predictions of working conditions 1 and 5, the time of arrival and departure of the oil slick at the water intake under the southwest wind condition is 2 min earlier than under the northeast wind condition; thus, when the oil spill accident occurs in the same flow field with the same wind speed, the direction of the wind has less influence on the speed of oil slick drift and spread. While comparing the predictions for operating conditions 2 and 5, the arrival and departure times of the oil slick are different. It can be seen that when the oil spill accident occurs in the same flow field and the same wind direction, the wind speed has a greater influence on the speed of oil slick drift and spread.

*3.2. Oil Spill Diffusion Results*

If an oil spill occurs at the wharf at high tide, the leaked petroleum-type pollutants will drift upstream of the wharf and will not pose a threat to the water quality of the proposed water outlet. in the event of an oil spill during the falling current, the leaked petroleum pollutants will spread to the proposed water outlet, which may affect the quality of the water intake. therefore, it makes sense to only consider the risk and impact of an oil spill at a shipping terminal at low tide.

In Figures 4 and 5, the black spot represents the size of the oil slick, i.e., the extent affected by the oil spill. The prediction results of the oil slick drift extension model show that when a leakage accident occurs during the dry period and a falling current, the oil slick moves upstream by the action of the current and drifts northeast by the action of the southwest wind. Under these conditions, it will take 46 min from the beginning of the spill for the oil slick front to reach the water intake of the drinking water source (Figure 4). The oil slick (the black spot) length is about 405 m and the width is about 270 m after passing through the water source. After 66 min, the oil slick drifts past the water intake (Figure 5). The length of the oil slick is about 465 m, the width is about 225 m, and the influence time on the water intake of the water plant is about 20 min.

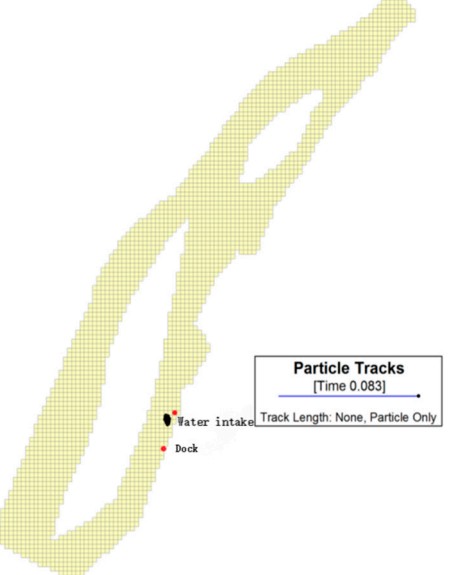

**Figure 4.** Influence range of oil slick drift in an oil spill accident (when the oil slick reaches the water intake).

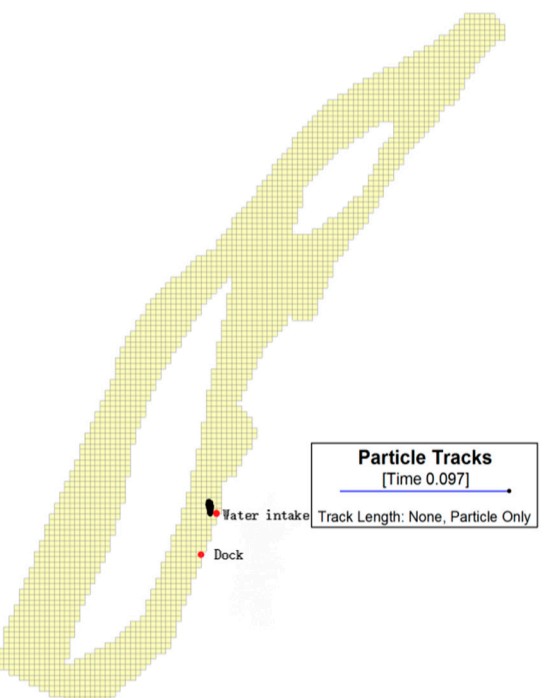

**Figure 5.** Influence range of oil slick drift in an oil spill accident (when the oil slick drifts out of the water intake).

It can be seen from the prediction results that, under the design's hydrological conditions, once accidental leakage occurs, because the water intake is close to the wharf, it will affect the water intake of the water plant sooner and the water quality of the intake will be greatly affected. However, the duration of the impact is relatively short, and the influence time on the proposed water outlet is about 20 min. The impact is greatly weakened after the floating oil slick passes through the water intake section.

### 3.3. Oil Spill Risk Assessment of Terminal Accidents

An environmental risk assessment of an oil spill quantitatively describes the possibility or uncertainty of environmental risk events and losses [12,39,40]. It has three parts: risk assessment, effectiveness evaluation of the control mechanisms, and vulnerability evaluation of receptors. Therefore, the structure of the risk evaluation index system for this project can be determined from these three aspects.

#### 3.3.1. Determination of the Risk Indicator System

In the case of sudden oil spillage in a river, there are some main factors related to the degree of impact of the accident [25,41].

(1) Leakage location: The closer the location of the oil spill accident is to the sensitive area, the higher the urgency of the accident.
(2) Nature of oil leakage: The toxicity, persistence, flammability, and other physical and chemical properties of spilled oil will have a direct impact on the degree of harm caused by the incident.
(3) Oil spillage: The total amount of oil leakage and leakage concentration will have an impact on the urgency and hazard of the accident.
(4) Leakage material form: The state (solid, liquid, or gaseous) of the leaked oil when it leaks has an impact on its diffusion and migration.
(5) Time of the incident: When the incident occurs—either during the day or night or during normal working hours or non-working hours, etc.—has an impact on the timeliness of the emergency response after the incident.

(6)     Accident form: Different accident forms (cabin explosion, breakage, tipping, etc.) will lead to differences in the mode of pollution (continuous leakage or instantaneous leakage), which will affect the scope and intensity of oil pollution in the water body.

(7)     Hydrological conditions: The flow rate and flow of the water body at the time of the incident play a decisive role in the diffusion and migration rates of oil products, and the flow direction has a direct impact on the diffusion and migration directions of oil products.

(8)     Weather conditions: The wind speed and direction at the time of the incident have an impact on the drift speed and direction of pollutants in the water, and the temperature plays a role in chemical changes such as volatilization and degradation of pollutants. Visibility will also have an impact on emergency response actions.

### 3.3.2. Establishment of the Risk Evaluation Index System

In the comprehensive analysis of the characteristics of sudden oil spillage in the evaluation area and the characteristics of oil movement changes in water, five indicators were proposed: oil nature, oil spillage, oil slick arrival time, emergency treatment measures, and receptor sensitivity [42–44].

Based on the comprehensive analysis of various factors that play a decisive role in the degree of influence of the consequences of a river oil spill accident and the possibility and difficulty of data acquisition, a river oil spill accident risk evaluation index system was determined.

The three elements of the environmental risk system—namely, risk source, control mechanism, and receptor—correspond to a total of five influencing factors. Among them, the risk source factors include two influencing factors: oil properties and oil spillage; the elements of the control mechanism include two influencing factors: oil slick arrival time and emergency disposal capacity; and the receptor element includes one influencing factor: sensitivity.

### 3.3.3. Classification of Early Warning Thresholds for Key Risk Indicators

(1)     Risk sources

(i)      Oil properties: toxicity and durability of oil spills

The hazardous characteristics of oils are mainly toxicity, durability, and flammability [45]. The core pollution receptor studied in this paper is the water quality of the water intake, so the flammability of the oil does not need to be considered for the time being. Oil hazards can be divided into five grades, of which the most harmful are "highly toxic and persistent" oils and "toxic and persistent" oils, mainly referring to some highly toxic originals, followed by "highly toxic and prominent" oils (gasoline, light kerosene, and other oils containing more aromatic hydrocarbons), "toxic and generally volatile oils" (heavy kerosene, heavy oils containing less aromatic hydrocarbons), and "low toxicity oils" (almost insoluble in water, heavy oils without aromatics, etc.).

The maximum score corresponding to the hazard index is 5, the minimum score is 1, and there are five levels, as detailed in Table 3.

**Table 3.** Oil hazard index score.

| Oil Hazards | Metric Score |
|---|---|
| Highly toxic and persistent (certain crude oils, etc.) | 5 |
| Toxic and persistent (general fuel oil, etc.) | 4 |
| Highly toxic and prominent (gasoline, light kerosene, and other oils containing more aromatics) | 3 |
| Toxic and generally volatile (heavy oil with less aromatic hydrocarbons, etc.) | 2 |
| Low toxicity (almost insoluble in water, heavy oils without aromatics, etc.) | 1 |

(ii)     Oil spillage

According to the actual shipping situation in this section and the statistical data of ship leakage accidents over the years, the leakage interval distribution law is analyzed, and the leakage level is divided into five levels, of which a leakage greater than 100 tons is defined as a super-large oil spill with a score of 5, and a micro oil spill event is defined as less than 0.1 tons with a score of 1. The index scores corresponding to the specific intervals are shown in Table 4.

**Table 4.** Oil spill index scores.

| Oil Spill (t) | >100 | 100~10 | 10~1 | 1~0.1 | <0.1 |
|---|---|---|---|---|---|
| Metric score | 5 | 4 | 3 | 2 | 1 |

(2) Control mechanism

(i) Oil slick arrival time

Through data research and expert consultation, according to the general evaluation results of the emergency response time of the water intake in response to sudden oil spillage, the oil slick arrival time index is divided into five categories: more than 12 h (very low risk), 6–12 h (low risk), 1–6 h (medium risk), 0.5–1 h (high risk), and less than 0.5 h (very high risk), as shown in Table 5.

**Table 5.** Oil slick arrival time index score.

| Arrival Time (h) | <0.5 | 0.5~1.0 | 1.0~6 | 6~12 | >12 |
|---|---|---|---|---|---|
| Metric score | 5 | 4 | 3 | 2 | 1 |

(ii) Accident emergency response capability

Perfect oil spill emergency equipment and facilities are available: oil spill emergency equipment is the hardware required to deal with sudden oil spills in a timely manner, such as oil pollution recovery ships, emergency equipment transport vehicles, etc.

Relevant emergency response capabilities are available: The construction of emergency capacity software cannot be ignored, which is crucial for determining whether oil spills can be dealt with in a timely and effective manner. Important emergency response capability also includes the management of staff, the formation and drilling of emergency prevention teams, and a smooth emergency communication network. Additionally, it is important to be able to formulate countermeasures for the disposal of oil pollution. These countermeasures could include the use of oil booms to contain and control oil pollutants on the water surface to prevent the expansion of the pollution area; the use of skimmers, adsorbing materials, dispersants, and other facilities to remove oil pollution from the water surface and water bodies; and the storage of the recovered oil dirt in special oil bladders and the cargo holds of bulk carriers and its processing at professional recycling units on shore to avoid secondary pollution.

According to the above standards for emergency response capacity, the emergency response capacity score is determined through a comprehensive evaluation of field research (Table 6).

**Table 6.** Emergency response capacity score.

| Ability | Weaker | Weak | Ordinary | Good | Better |
|---|---|---|---|---|---|
| Metric score | 5 | 4 | 3 | 2 | 1 |

(3) Receptor sensitivity

According to the "Functional Zoning of Surface Water Environment in Jiangsu Province" released by the People's Government of Jiangsu Province, the water quality indicators of the receptor are divided into five levels, as shown in Table 7.

**Table 7.** Water quality index scores.

| Water Quality | I | II | III | IV | V |
|:---:|:---:|:---:|:---:|:---:|:---:|
| Metric score | 5 | 4 | 3 | 2 | 1 |

### 3.3.4. Indicator Weight

Expert scoring is used to determine the weighting of each metric. The 12 experts scored the indicators of each column with a full score of 100. Then the indicators were preprocessed, weighted, and summed to obtain the final score. The percentages were then calculated to obtain the corresponding weight of each index.The specific evaluation index system and its weights are shown in Table 8.

**Table 8.** Risk evaluation index system and weights of sudden oil spill accidents.

| A | B | A-B Weight | C | B-C Weight | A-C Weight |
|:---:|:---:|:---:|:---:|:---:|:---:|
| | Risk sources | 0.491 | Oil properties | 0.29 | 0.142 |
| | | | Oil spillage | 0.71 | 0.349 |
| Risk Evaluation Composite Index | Control mechanism | 0.358 | Oil slick arrival time | 0.68 | 0.243 |
| | | | Emergency response measures | 0.32 | 0.115 |
| | Receptor | 0.151 | Sensitivity | 1 | 0.151 |

*3.4. Comprehensive Evaluation of the Calculation and Grading of Consequences*

### 3.4.1. Calculation of Consequences

Based on the above requirements, the actual data is collected, the values of each evaluation index are summarized, and the evaluation consequences are calculated as follows:

$$\text{I} = \sum_{i=1}^{5} w_{c_i} k_{c_i} \tag{11}$$

where I represent the comprehensive evaluation index; $w_{c_i}$ indicates the composite weight of the evaluation index $c_i$ for the target layer A; and $k_{c_i}$ indicates the evaluation score obtained by the evaluation index $c_i$ based on the actual statistical data reference index evaluation standard.

### 3.4.2. Comprehensive Evaluation Index Grading

According to the actual shipping situation described in this section, the historical data of ship leakage accidents over the years in this area, and the model trial results, the consequences of oil spill risk assessment are divided into five levels (Table 9).

**Table 9.** Classification of consequences of oil spill risk assessment.

| I Value | >3.51 | 3.51~3.26 | 3.26~3.01 | 3.01~2.53 | <2.53 |
|:---:|:---:|:---:|:---:|:---:|:---:|
| Warning level | V | IV | III | II | I |

When a sudden oil spill occurs near the water intake of a water source/plant, the principle of "life before production" is adopted on the issue of water supply, and ensuring the safety of the domestic water supply is given priority. Specifically, actions are taken with regard to the risk level of a sudden oil spillage accident to the water intake of the water plant. The emergency classification mechanism of an accident is as follows:

(1)　Risk level I is extremely low. Oil spills have fewer oil substances leaked, which do not pose a threat to the normal intake of water and the water supply.

(2)　Risk level II is low risk. The location of the oil spill accident is a certain distance from the water body or occurs near the water, but the oil pollution has not yet reached the entire water body or only a small part of it, and the degree of harm is not large. By strengthening conventional processes and emergency treatment technical measures, the production of the water plant is not reduced or moderately reduced.

(3)　Risk level III is medium risk. That is, a large amount of oil pollution has leaked partially or completely into the body of water, and the degree of harm is relatively large. By strengthening the conventional processes, the water supply of the water plant can be kept at 70–80%.

(4)　Risk level IV is medium to high risk. A large amount of oil pollution has entered the water body as a whole, and the degree of harm is large. By strengthening conventional processes and taking emergency treatment technical measures, the water supply of the water plant can be kept at 50–60%.

(5)　Risk level V is high risk. That is, the amount of pollutants is large and has entered the water body and spread, posing a direct threat to the safety of the water intake. By strengthening conventional processes and emergency treatment technical measures, the water supply of the water plant is reduced to 50%, and if the pollutant indicators in the source water seriously exceed the standard and the emergency treatment technical measures are still difficult to deal with, measures to suspend the production and supply of tap water should be taken after consulting the local government. If water pollution occurs near the water source of the downstream waterworks, countermeasures will be taken to reduce the water supply of the downstream water plant and increase the water supply of the upstream water plant.

According to accident information (Table 10), scores of accident risk indicators (Table 11) and the risk evaluation index system and weights for sudden oil spill accidents (Table 8), the comprehensive evaluation index of drinking water sources is calculated at 3.225. According to classification of consequences of oil spill risk assessment (Table 9), under the current emergency measures of the shipping terminal, the level of risk associated with the occurrence of a ship collision oil spill accident at the drinking water source is level III (medium risk).

**Table 10.** Accident information.

| Incident Information Item | Content |
|---|---|
| Place | Dock |
| Time | Low tide period |
| Risk receptors | Water intakes at the source of drinking water |
| Oil | Light diesel |
| Oil spillage | 10 t |

**Table 11.** Scores of accident risk indicators.

| Evaluate Metrics | Accident Records | | Metric Score |
|---|---|---|---|
| Oil properties | diesel oil | | 1 |
| Oil spillage | 10 t | | 3 |
| Arrival time | Drinking water sources | 46 min | 4 |
| Emergency response measures | Dock | weak | 4 |
| Receptor | Drinking water sources | I | 4 |

## 4. Conclusions

It can be seen from the prediction results that, under the hydrological design conditions of the model, once an accidental leakage occurs, the oil spill at a wharf will soon have an impact on a water plant's water intake due to the closeness of the water intake to that wharf. If accidental emergency measures are needed, it is best if the impact duration of the accident is relatively short, and the impact time on the water inlet is about 20 min. After the floating oil slick passes through the water inlet section, the impact is greatly weakened.

In order to protect the quality of drinking water sources, strict environmental management must be adopted to prevent such accidents as much as possible. Our work supports establishing relevant preventive and emergency systems, improving equipment, improving personnel quality, and formulating oil spill contingency plans. We feel that adopting these action items will help prevent oil spill accidents. Once a risk accident occurs at a shipping terminal, an oil spill emergency plan should be activated immediately, and emergency measures should be taken to reduce the impact of oil spill accidents on the water environment.

**Author Contributions:** Conceptualization, F.H. and J.M.; Data curation, J.S.; Formal analysis, Q.L.; Investigation, F.H. and J.M.; Methodology, F.H.; Supervision, W.L.; Visualization, J.M.; Writing—original draft, J.M.; Writing—review & editing, F.H. All authors have read and agreed to the published version of the manuscript.

**Funding:** This research was funded by The Project of Ecological and Environmental Protection Integration Research Institute in the Yangtze River Delta (No. ZX2022QT046). Major Science and Technology Program for Water Pollution Control and Treatment (No. 2017ZX07301006) and The Special Fund of Chinese Central Government for Basic Scientific Research Operations in Commonweal Research Institute (No. GYZX220405).

**Data Availability Statement:** The data used during the study appear in the submitted article.

**Conflicts of Interest:** The authors declare no conflict of interest. The funders had no role in the design of the study; in the collection, analyses, or interpretation of data; in the writing of the manuscript, or in the decision to publish the results.

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
