# Peer review of "Environmental Impact Assessment of a Wharf Oil Spill Emergency on a River Water Source"

_water, doi:10.3390/w15020346_

Round 1

Reviewer 1 Report

Review of "Environmental Impact Assessment of a Wharf Oil Spill Emergency on a River Water Source".

General assessment:

In this study, a risk assessment scheme for oil spill pollution is used to evaluate the sudden pollution incident near a water intake in Changjiang river. In the scheme, the oil spill model is used to predict the behavioral processes of the oil spilled from dock, and the analytic hierarchy process is used to do the risk assessment. The manuscript is suitable for this journal, but there are several aspects to be improved. So, I think the manuscript needs a major revision. If the details cannot be clarified, it may be needed to reject the manuscript.

Comments:

(1) The hydrodynamic condition is critical for the predicting the transport of the spilled oil, which is done by the ‘two-dimensional power flow model’ in this study. The model must be evaluated by the observations. If not, the arrival time of 46 min obtained in this study may be unbelievable.

(2) Except for the flow, the wind and temperature can also affect the behavioral processes of the spilled oil, are they considered in this study? As shown in Equation 7, is the effect of the wind considered in Ucx? If so, how to calculate it? In line 216, ‘the wind speed is 5.0 m/s after frequency analysis’, how to do the frequency analysis and what is the wind direction? How to calculate the variation of the temperature in the river in the study?

(3) In the analytic hierarchy process, the indicator weight is very important. As shown in line 367, ‘Expert scoring is used to determine the weighting of each metric’, which is usually used. But the details are not described. How many questionnaires filled by the experts are used? How to calculate the weighting? Is the significance testing used during the calculation of the weighting?

(4) Many details are not described in this study.

For example, what is the location of the study area? From Figure1 and the text, I know it is a part of the Changjiang River, but which part? What is the longitude and latitude and the location of the study area? Where is the upstream and downstream?

For example, in lines 141-147, ‘this numerical method is divided into divided into several different sub-operator equations according to the different characteristics of the operators contained in the equation, and the equations of each sub-operator can be solved by the corresponding numerical methods.’ How to divide the numerical method and what is the corresponding numerical methods?

As described in lines 148-151, the roughness is spatially varying. What is the spatial distribution of the roughness (n in line 131) used in the simulation in this study?

In lines 379-381, what is the historical data?

In line 363, what is ‘Functional Zoning of Surface Water Environment’?

(5) There are many wrong expressions in the manuscript. It is recommended to polish language by the other researcher and professional English editor.

For example, in line 88, ‘tons’ is used. But in line 91, ‘t’ is used.

In line 120, ‘power flow model’ is rarely used in this field.

In line 215, ‘Maxus Station’ is error. It should be ‘Datong Station’.

In lines 219-220, ‘half-day tide’ and ‘daily tide’ are rarely used in this field.

In lines 314-322, the descriptions of the five grades of oil properties are not consistent with those listed in Table 2.

In line 339, ‘612 hours’, ’16 hours’, and ‘0.51 hour’ are error. According to Table 4, it may be ‘6-12 hours’, ’1-6 hours’ and ‘0.5-1 hour’.

Author Response

List of Responses

Thank you for your letter and for the reviewers’ comments concerning our manuscript entitled “Environmental Impact Assessment of a Wharf Oil Spill Emergency on a River Water Source”. Those comments are all valuable and very helpful for revising and improving our paper, as well as the important guiding significance to our research. Taking account of reviewers’ comments, we have revised and improved the manuscript. We hope our revisions meet with approval. Revised portion is marked up using the “Track Changes” function in the paper. The main corrections in the paper and the responses to the reviewers’ comments are as follows.

Reviewer 1

Comments and Suggestions for Authors:

In this study, a risk assessment scheme for oil spill pollution is used to evaluate the sudden pollution incident near a water intake in Changjiang river. In the scheme, the oil spill model is used to predict the behavioral processes of the oil spilled from dock, and the analytic hierarchy process is used to do the risk assessment. The manuscript is suitable for this journal, but there are several aspects to be improved. So, I think the manuscript needs a major revision. If the details cannot be clarified, it may be needed to reject the manuscript.

Point 1: The hydrodynamic condition is critical for the predicting the transport of the spilled oil, which is done by the ‘two-dimensional power flow model’ in this study. The model must be evaluated by the observations. If not, the arrival time of 46 min obtained in this study may be unbelievable.

Response 1: Thanks for the reviewer’s suggestion. We are very sorry for our incorrect wroding and the misunderstanding we caused.The the arrival time of 46 min obtained in this study was from the prediction results of the oil film drift extension model. The process was explained on page 8, line 256-268.

Point 2: Except for the flow, the wind and temperature can also affect the behavioral processes of the spilled oil, are they considered in this study? As shown in Equation 7, is the effect of the wind considered in Ucx? If so, how to calculate it? In line 216, ‘the wind speed is 5.0 m/s after frequency analysis’, how to do the frequency analysis and what is the wind direction? How to calculate the variation of the temperature in the river in the study?

Response 2: Thanks for the reviewer’s suggestion. We have increased more statement about the wind and temperature that affect the behavioral processes of the spilled oil on page 6, line 227-232 of the revised version.

The wharf is located in the southwest direction of the water intake, so we choose the southwest wind, the most unfavorable wind, as the wind direction. And the wind speed is the instantaneous maximum wind speed of the southwest wind with 5.0 m/s. The climate of the study area belongs to the northern subtropical monsoon climate zone, with an annual average temperature of 15 ℃, so we choose 15 ℃ as the tem-perature.

Point 3: In the analytic hierarchy process, the indicator weight is very important. As shown in line 367, ‘Expert scoring is used to determine the weighting of each metric’, which is usually used. But the details are not described. How many questionnaires filled by the experts are used? How to calculate the weighting? Is the significance testing used during the calculation of the weighting?

Response 3: Thanks for the reviewer’s suggestion. We have increased more details about the expert scoring we used in determining the weighting of each metric on page 12, line 386-389 of the revised version.

The 12 experts scored the indicators of each column with a full score of 100. Then the indicators were preprocessed, weighted and summed to obtain the final score. Then calculate the percentage to get the corresponding weight of each index.

Point 4: Many details are not described in this study.

For example, what is the location of the study area? From Figure1 and the text, I know it is a part of the Changjiang River, but which part? What is the longitude and latitude and the location of the study area? Where is the upstream and downstream?

For example, in lines 141-147, ‘this numerical method is divided into divided into several different sub-operator equations according to the different characteristics of the operators contained in the equation, and the equations of each sub-operator can be solved by the corresponding numerical methods.’ How to divide the numerical method and what is the corresponding numerical methods?

As described in lines 148-151, the roughness is spatially varying. What is the spatial distribution of the roughness (n in line 131) used in the simulation in this study?

In lines 379-381, what is the historical data?

In line 363, what is ‘Functional Zoning of Surface Water Environment’?

Response 4: Thanks for the reviewer’s suggestion. Again, we are very sorry for our negligence of the explanation. First, we have added more information about the study area attached with a figure about the relative position of Dajian Wharf and the water intake on page 2, line 74-81 of the revised version. Then we have modified the woding of roughness on page 5, line 160-163 of the revised version, the content on page 11, line 381-382, page 12, line 402 of the revised version.

(1) Nanjing Dajian Wharf (118°32′ E, 24°52′ N) and Jiangning Binjiang Water intake (118°32 E, 31°50′ N) are both located in Tongjing Port Area of Nanjing Jiangning Bin-jiang Economic Development Zone. They are located in the western part of Jiangning District of Nanjing, the capital of Jiangsu Province, the central city of the Yangtze Riv-er Delta. They are close to the golden waterway of the Yangtze River, 25 km from the main city of Nanjing, about 40 km from the downstream Nanjing Xinshengwei Port Area, about 54 km from the upstream Wuhu Port, and about 432 km from Shanghai Port.

Point 5: There are many wrong expressions in the manuscript. It is recommended to polish language by the other researcher and professional English editor.

For example, in line 88, ‘tons’ is used. But in line 91, ‘t’ is used.

In line 120, ‘power flow model’ is rarely used in this field.

In line 215, ‘Maxus Station’ is error. It should be ‘Datong Station’.

In lines 219-220, ‘half-day tide’ and ‘daily tide’ are rarely used in this field.

In lines 314-322, the descriptions of the five grades of oil properties are not consistent with those listed in Table 2.

In line 339, ‘612 hours’, ’16 hours’, and ‘0.51 hour’ are error. According to Table 4, it may be ‘6-12 hours’, ’1-6 hours’ and ‘0.5-1 hour’.

Response 5: Thanks for the reviewer’s suggestion. We regret there were problems with the English. The paper has been carefully revised by a professional language editing service to improve the grammar and readability. The wrong expressions in the manuscript are all revised on page 3, line 101, page 4, line 130, page 6, line 226, page 6, line 235, page 10, line 335-337, page 11, line 357 of the revised version.

Attached is the certificate of language polishing.

Reviewer 2 Report

The article presents an interesting methodology to assess the impact of oil spills on rivers and waterways. Simulation results and risk analysis can guide managers about the risks of oil spill accidents.

I recommend authors check the line  339, because some data were not corresponding to the correct values ​​from Table 4 - *612 hours (low risk), *16 hours (medium risk), 0.51 hours (high risk), and less than 0.5 hours (very high risk), as shown in Table 4.

So, I recommend that this article be accepted for publication.

Author Response

List of Responses

Thank you for your letter and for the reviewers’ comments concerning our manuscript entitled “Environmental Impact Assessment of a Wharf Oil Spill Emergency on a River Water Source”.  Again, we are very thankful for your recognition of the article

Reviewer 2

Comments and Suggestions for Authors:

The article presents an interesting methodology to assess the impact of oil spills on rivers and waterways. Simulation results and risk analysis can guide managers about the risks of oil spill accidents.

I recommend authors check the line 339, because some data were not corresponding to the correct values ​​from Table 4 - *612 hours (low risk), *16 hours (medium risk), 0.51 hours (high risk), and less than 0.5 hours (very high risk), as shown in Table 4.

Response: Thanks for the reviewer’s suggestion and the recommendation that this article be accepted for publication. We regret there were problems with the English. The wrong expressions in the manuscript are all revised on page 11, line 357 of the revised version.

Round 2

Reviewer 1 Report

(1) I know the 46 min was obtained by the oil spilled model. If the hydrodynamic condition is not evaluated, the simulated results of the oil spilled model are unbelievable. Please evaluate the hydrodynamic condition using the hydrodynamic observations, or demonstrate that the simulated results of the hydrodynamic model and oil spilled model are believable.

(2) How to calculate the effect of the wind and temperature in the oil film drift extension model? Please give the details of the oil film drift extension model.

(3) Please combine the Figure1 and Figure2, i.e., show the calculation area in Figure1.

Author Response

Thank you again for your letter and for the reviewers’ comments concerning our manuscript entitled “Environmental Impact Assessment of a Wharf Oil Spill Emergency on a River Water Source”. Those comments are all valuable and very helpful for revising and improving our paper, as well as the important guiding significance to our research. Taking account of reviewers’ comments, we have revised and improved the manuscript. We hope our revisions meet with approval. Revised portion is marked up using the “Track Changes” function in the paper. The main corrections in the paper and the responses to the reviewers’ comments are as follows.

Reviewer 1

Comments and Suggestions for Authors:

Point 1: (1) I know the 46 min was obtained by the oil spilled model. If the hydrodynamic condition is not evaluated, the simulated results of the oil spilled model are unbelievable. Please evaluate the hydrodynamic condition using the hydrodynamic observations, or demonstrate that the simulated results of the hydrodynamic model and oil spilled model are believable.

Response 1: Thanks for the reviewer’s suggestion. We have increased more statement about the model rate validation to describe reliability of the hydrodynamic condition we chose on page 6, line 233-249 of the revised version.

  • The overall one-dimensional water flow and local two-dimensional tidal current mathematical model were used to simulate the Jiangsu section of the Yangtze River. According to the division of the two-dimensional simulation calculation area, the one-dimensional model provides the corresponding water flow boundary conditions for the two-dimensional model and then the two-dimensional regional water quality coupling numerical simulation is carried out.
  • Datong Station with measured data were used as test sections, and the measured water level and the simulated calculated water level were compared. The data from 6:00 a.m. on January 14, 1979, to 6:00 a.m. on January 18, 1979 (96 hours in total) were used for model calibration, and the measured water level and the simulated calculated water level were compared respectively.The results show that the water surface line along the course is basically the same as the measured water surface line, and the percentage of the time period with the tidal level error not exceeding 20 cm in the total rate fixed period is between 80% and 98%. In general, the generalization of the area by the one-dimensional water flow model is basically reasonable, and the parameters selected basically reflect the hydraulic characteristics of the river.

Point 2: How to calculate the effect of the wind and temperature in the oil film drift extension model? Please give the details of the oil film drift extension model.

Response 2: Thanks for the reviewer’s suggestion. We have added more description about the effect of the wind in the oil film drift extension model on page 6, line 227-230, page 8, line 278-292 of the revised version.

  • The influence of temperature on oil spill evaporation involves the evaporation rate and total evaporation ratio; the higher the temperature, the faster the oil evaporates; for the same oil, the total evaporation ratio is large at high temperatures and small at low temperatures. While the temperature has less impact in the oil film drift extension model.
  • Because the oil film drift process mainly depends on the action of the wind field and flow field, it is necessary to study the influence of the wind field on oil film drift trajectory. The predicted results of working conditions 1, 2, and 3 were compared when all else was equal, and the effects of wind direction and speed on the drift and diffusion of the oil film were analyzed.
  • Comparing the predictions of working conditions 1 and 5, the time of arrival and departure of the oil film at the water intake under the southwest wind condition is 2 minutes earlier than under the northeast wind condition. It can be seen that when the oil spill accident occurs in the same flow field with the same wind speed, the direction of the wind has less influence on the speed of oil film drift and spread. While comparing the predictions for operating conditions 2 and 5, the arrival and departure times of the oil film are different. It can be seen that when the oil spill accident occurs in the same flow field and the same wind direction, the wind speed has a greater influence on the speed of oil film drift and spread.

Point 3: Please combine the Figure1 and Figure2, i.e., show the calculation area in Figure1.

Response 3: Thanks for the reviewer’s suggestion. We have combined the Figure1 and Figure2 on page 3, line 84 of the revised version.
